# Unstable Limit Cycles and Singular Attractors in a Two-Dimensional Memristor-Based Dynamic System

**DOI:** 10.3390/e21040415

**Published:** 2019-04-19

**Authors:** Hui Chang, Qinghai Song, Yuxia Li, Zhen Wang, Guanrong Chen

**Affiliations:** 1College of Electrical Engineering and Automation, Shandong University of Science and Technology, Qingdao 266590, China; 2College of Mathematics and Systems Science, Shandong University of Science and Technology, Qingdao 266590, China; 3Department of Electronic Engineering, City University of Hong Kong, Hong Kong, China

**Keywords:** memristor, hopf bifurcation, unstable limit cycle, singular attractor, coexistence of attractors

## Abstract

This paper reports the finding of unstable limit cycles and singular attractors in a two-dimensional dynamical system consisting of an inductor and a bistable bi-local active memristor. Inspired by the idea of nested intervals theorem, a new programmable scheme for finding unstable limit cycles is proposed, and its feasibility is verified by numerical simulations. The unstable limit cycles and their evolution laws in the memristor-based dynamic system are found from two subcritical Hopf bifurcation domains, which are subdomains of twin local activity domains of the memristor. Coexisting singular attractors are discovered in the twin local activity domains, apart from the two corresponding subcritical Hopf bifurcation domains. Of particular interest is the coexistence of a singular attractor and a period-2 or period-3 attractor, observed in numerical simulations.

## 1. Introduction

Memristors postulated by Chua in 1971 [1] have gradually attracted increasing attention from scientific and technological communities. Thereafter, various memristors were mathematically modeled and physically fabricated. Generally, memristors can be divided into four categories, namely ideal, ideal generic, generic, and extended memristors [2,3]. The first ever memristor is an ideal device [1], and the one built by HP Labs in 2008 is an ideal generic prototype [4]. Both ideal and ideal generic memristors are widely used as new components in neural networks [5,6,7], neural synapses [8], and chaotic circuits [9,10,11,12,13]. In addition, memristors are introduced into oscillating circuits due to their inherent nonlinearity to generate pseudo-random signals, which can be used for secure communication and image encryption [14,15,16]. In virtue of the ultra-low energy consumption and fast switching speed, they also gain popularity in many new computer architectures [17]. Some generic memristors in such as Hodgkin–Huxley Axon memristor model [18,19], and Morris–Lecar memristor model [20,21] have been identified through detailed and systematic qualitative analyses, which contribute to the penetrate complex dynamics of biological membranes. The Chua shoelace memristor [2], named as the Chua corsage memristor in [22,23], is also a generic memristor and possesses only one local activity domain, which presages the existence of complex dynamics in associated memristor systems [24]. An oscillator system based on this kind of memristor can have periodic solutions, particularly stable limit cycles can be found in its supercritical Hopf bifurcation domain. It has been noticed that intrinsic nonlinear characteristics are valuable and significant for understanding the operational mechanisms of memristors.

A generic memristor, possessing coexisting pinched hysteresis loops and twin local-activity domains, is designed and named a bistable bi-local active memristor (BBAM) in [25]. Here, bistability means that the memristor has two different stable solutions originating from both sides of the critical value of the initial state under the same input control; bi-local activity means that the memristor has two bi-local active domains; the memristor with two distinct stable pinched hysteresis loops and two local activity domains is referred to as a bistable bi-local-active memristor (see [25] for more details). Since the local activity theory provides a coherent framework for the complex dynamics of memristor-based circuit systems, it is an interesting topic to further explore the dynamical behaviors in twin local-activity domains of BBAM. The circuit based on the memristor is given, inspired by [3,22], as shown in Figure 1.

The corresponding dynamic system is described by the following two-dimensional system:
(1a)$$\begin{array}{c} \frac{dx}{dt}=-x\left|x\right|+5x+\frac{i_{L}}{x^{2}}, \end{array}$$
(1b)$$\begin{array}{c} \frac{di_{L}}{dt}=\frac{1}{L}\left(V-\frac{i_{L}}{x^{2}}\right), \end{array}$$
where x=x(t) is the internal state variable, and $i_{L}=i_{L}\left(t\right)$ is the current through the inductor, with inductance *L*. Its mathematical model is shown inside the rectangle box of Figure 1, where x=x(t) represents the memristor internal state, V=v(t) represents an external input signal and i=i(t) represents an output signal. This memristor has twin local-activity domains, as shown in Table 1.

The twin subcritical Hopf bifurcation domains of this system having unstable limit cycles are shown in Table 2. Technically, there are some difficulties in finding unstable limit cycles by using the available schemes (e.g., [21]). Thus, how to find available unstable limit cycles quickly is an important and interesting issue for further exploration.

The paper aims is to present an easy operational scheme for finding unstable limit cycles and to explore the complex dynamical behaviors of system (1). More precisely, in this paper, unstable limit cycles will be found by the designed algorithm and their evolution law will be revealed by numerical analysis, and moreover coexisting singular attractors will be identified in the local activity domains of the memristor.

The rest of the paper is organized as follows. A programmable scheme for finding unstable limit cycles is designed and verified by numerical simulations in Section 2. In Section 3, some coexisting singular attractors are identified in the twin local active domains of the memristor, apart from the corresponding subcritical Hopf bifurcation domains. The last section presents some concluding remarks.

## 2. Unstable Limit Cycles of the Memristor-Based Dynamic System

In order to find unstable limit cycles of a dynamic system, some feasible schemes were suggested (e.g., [21]). However, the available algorithms are either difficult to program or time-consuming to operate when applied to system (1). Therefore, it is desirable to develop an easy-to-operate scheme, which is the objective of this section.

The proposed programmable scheme searching for unstable limit cycles is shown in Figure 2. Let the two equilibrium points of the system be $Q_{0}$ and $Q_{2}$, which are located inside the twin subcritical Hopf bifurcation domains where an unstable limit cycle exists, respectively, as specified in Table 2. Assume that the system state tends to $Q_{0}$ (for $Q_{2}$ the process is similar).

**Remark** **1.**The disturbance terms $\varepsilon_{0}$, $\delta_{0}>0$, with $\varepsilon_{0}<\delta_{0}$ and $a_{i}<b_{i}$, i, j=0,1,2,…, are constants through the whole process. At Step 4, if the state converges to $Q_{2}$ instead, another pair of points can be chosen, where $A_{0}$ is marked as $A_{1}\left(x_{0},a_{1}\right)$ and $C_{0}$ is marked as $B_{1}\left(x_{0},b_{1}\right))$, and the state starting from $A_{1}$ (or $B_{1}$) tends to $Q_{0}$ (or $Q_{2}$), respectively. In this case, the scheme could start with initial disturbance $\varepsilon_{0}<0$, $\delta_{0}<0$, and the procedure is similar.

Algorithm 1 is verified as follows. Letting the parameters V=5.996 V, 5.97 V, and 5.9605 V in the subcritical Hopf bifurcation domain: 5.96018 V <V<6 V, an unstable limit cycle of system (1) is found by using the above algorithm. Similarly, other ones are obtained when V=−5.996 V, −5.97 V, and −5.9605 V in the twin domain of −6 V <V<−5.96018 V. Specifically, when the applied DC voltages V=±5.996 V, and symmetrical initial conditions (−3,56.74833) and (3,−56.74833) about the origin are selected, so that one pair of unstable limit cycles with respect to the origin symmetrically emerge from system (1), which are both ellipses as shown in the two sub-graphs below Figure 3. In addition, the corresponding equilibrium points $Q_{0}\left(-3.00398,54.10743\right)$ and $Q_{0}^{*}\left(3.00398,-54.10743\right)$ are obtained, which are also symmetrical about the origin. Time domain waveforms of the system states are shown in two pairs of the upper and middle subgraphs of Figure 3. The initial conditions and applied DC voltages for generating them are listed in the upper parts of Figure 3a,b, respectively.

**Algorithm 1:** An algorithm for finding unstable limit cycles.
*Step 1*. Determine the equilibrium points $Q_{0}\left(x_{0},y_{0}\right)$ and $Q_{2}$ of the system.*Step 2*. Disturb the coordinate component $y_{0}$ of $Q_{0}$ with a smaller perturbation $\varepsilon_{0}$, while $x_{0}$ is fixed, so that the system state originating from $A_{0}\left(x_{0},a_{0}\right)$ moves towards $Q_{0}$, where $a_{0}=y_{0}+\varepsilon_{0}$.*Step 3*. Trigger a larger perturbation $\delta_{0}$ to $y_{0}$, so that the system state starting from $B_{0}\left(x_{0},b_{0}\right)$ tends to $Q_{2}$, where $b_{0}=y_{0}+\delta_{0}$.*Step 4*. Select the initial condition as the midpoint $C_{0}$
$\left(x_{0},\frac{a_{0}+b_{0}}{2}\right)$ between $A_{0}$ and $B_{0}$, if an unstable limit cycle emerges, the process terminates; if not, the system state converges to $Q_{0}$.*Step 5*. Mark $C_{0}$ as $A_{1}\left(x_{0},a_{1}\right)$ and mark $B_{0}$ as $B_{1}$
$\left(x_{0},b_{1}\right)$, where $a_{1}=\frac{a_{0}+b_{0}}{2}$ and $b_{1}=b_{0}$. Then, select the initial condition as the midpoint $C_{1}\left(x_{0},\frac{a_{1}+b_{1}}{2}\right)$ between $A_{1}$ And $B_{1}$. If an unstable limit cycle emerges, the process terminates; if not, the process continues, with $A_{2}$ And $B_{2}$, $A_{3}$ And $B_{3}$, …, until an unstable limit cycle emerges.

Similarly, when the voltages V=±5.97 V and V=±5.9605 V, the corresponding unstable limit circles and equilibrium points are symmetrical about the origin, as shown in Figure 4 and Figure 5.

Evolution law can be drawn from Figure 3a, Figure 4a, and Figure 5a, where one side of the unstable limit cycle gradually becomes narrower as the applied DC voltage *V* decreases in one subcritical Hopf bifurcation region, 5.96018 V <V<6 V. When the voltage *V* crosses the critical value $V^{*}$ (≈5.96018), the transient oscillation dies out, i.e., $V^{*}$ is a left-end point of the subcritical bifurcation region. Similarly, one side of the unstable limit cycle also becomes narrower as the voltage *V* increases in another region of −6 V <V<−5.96018 V, as shown in Figure 3b, Figure 4b, and Figure 5b. It is easy to verify that $V^{*}$ (≈−5.96018) is also a critical value to distinguish Hopf bifurcation from non-Hopf bifurcation.

## 3. Coexisting SingularAttractors of the Memristor-Based Dynamic System

Coexistence of attractors is often observed in higher-dimensional nonlinear autonomous dynamical systems [26,27,28] or a two-dimensional nonlinear non-autonomous system [29]. For two-dimensional autonomous systems, however, this phenomenon is generally not possible. Nevertheless, singular attractors and their coexistence were found from lower-dimensional memristor-based dynamical systems, in their local active domains, which is consistent with the prediction of the local active principle [24]. In fact, the existence of local active domains of the memristor betokens complex dynamics of the associated dynamical system, which theoretically confirms the existence of strange attractors. Some singular attractors are identified in the local active domains of −6 V <V<−4.6875 V and 4.6875 V <V<6 V of system (1), as further discussed below.

Letting the voltage V=−5.0296 V (or V=5.0296 V), the initial conditions x(0)=−3.605, i(0)=65.355 A (or x(0)=3.605, i(0)=−65.355 A), a singular attractor of system (1) is found, as shown on the right-hand side of Figure 6a (or Figure 6b). Its corresponding time-domain waveforms of variables x(t) and i(t) are displayed on the left-hand side of Figure 6a (or Figure 6b).

When V=±5.15 V, the singular attractors and their corresponding time-domain waveforms of the system states are shown in Figure 7a,b. In addition, the voltages V=±5.0296 V, ±5.15 V, and their corresponding initial conditions are marked at the upper left corners of Figure 6a,b and Figure 7a,b, respectively. It can be seen from Figure 6a,b that two singular attractors and two equilibrium points $Q_{0}$ and $Q_{0}^{*}$ are all symmetrical about the origin when the voltages *V* are selected as a pair of opposite values, V=±5.0296 V. This phenomenon also arises when the voltages V=±5.15 V, as shown in Figure 7.

Next, the question if coexisting attractors can emerge from system (1) is discussed. By selecting an appropriate voltage *V* from the twin local active domains of the memristor, coexisting attractors of the associated dynamical system can indeed be found by numerical simulations. Figure 8 shows the coexisting attractors and their evolutions with voltages V=±5.0354 V, with durations *t* = 0–300 s and *t* = 8700–9000 s, respectively. The initial condition and time length for each attractor are marked in Figure 8a–d. As time *t* increases, one of the two singular attractors (the red attractor) shown in Figure 8a (or Figure 8c) gradually becomes a period-2 attractor, as shown in Figure 8b (or Figure 8d). Figure 9 displays the evolutions of the coexisting attractors with voltages V=±5.18027 V and durations *t* = 0–300 s and *t* = 3700–4000 s. The initial conditions and time duration for each attractor are listed in Figure 9a–d. When the duration of the simulation is *t* = 3700–4000 s, one of the coexisting attractors (the red attractor) in Figure 9a (or Figure 9c) evolves to a period-3 attractor, as shown in Figure 9b (or Figure 9d). It can be seen from Figure 8 that one pair of attractors shown in Figure 8a and the corresponding attractors shown in Figure 8c are symmetrical about the origin, and the same can be seen in Figure 8b,d, Figure 9a,c, and Figure 9b,d. Subtle observations reveal that the initial conditions and parameters associated with the corresponding attractors are also symmetrical about the origin.

In summary, the coexistence of singular attractors indicates that, starting from different initial conditions, the system state will move to different regions and form different attractors under parameter *V* variations. It is interesting to find that when the parameter value of *V* is changed to the opposite value, the corresponding attractor will appear and is also symmetrical with respect to one in the original system. This is the external response of the intrinsic properties of the memristive twin local active domains in the associated dynamical system.

## 4. Conclusions

This paper presents a programmable scheme for finding unstable limit cycles and explores the complex dynamics of the associated dynamical system in the twin local-activity domains of the memristor. Unstable limit circles are found by the proposed algorithm, which appear in pairs and are symmetrical about the origin in the two subcritical Hopf bifurcation regions, determined by the intrinsic properties of the system. In the twin local activity domains of the memristor, coexisting singular attractors emerge from the associated dynamical system. This coexistence indicates that the system is very sensitive to initial conditions. Easily missed but not unimportant, the coexisting attractors also appear in pairs and are symmetrical about the origin, which is the external response of the symmetry of the bistable bi-local active memristor. The twin local active domains of the memristor present the complexity of its associated dynamical system, indicating that many other types of dynamical systems constituting of the memristor deserve further exploration. 

## Figures and Tables

**Figure 1 entropy-21-00415-f001:**
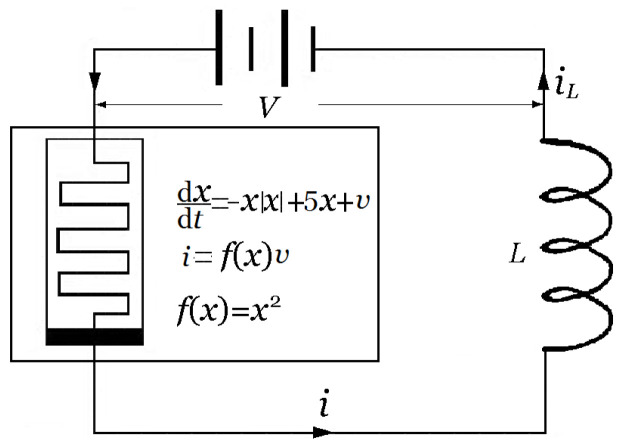
A circuit based on a memristor and an inductor with inductance L=1/27H [25].

**Figure 2 entropy-21-00415-f002:**
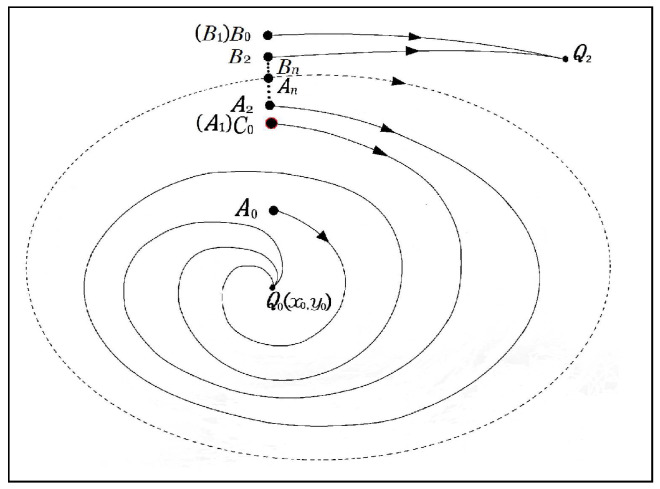
Process for finding an unstable limit cycle of a dynamic system.

**Figure 3 entropy-21-00415-f003:**
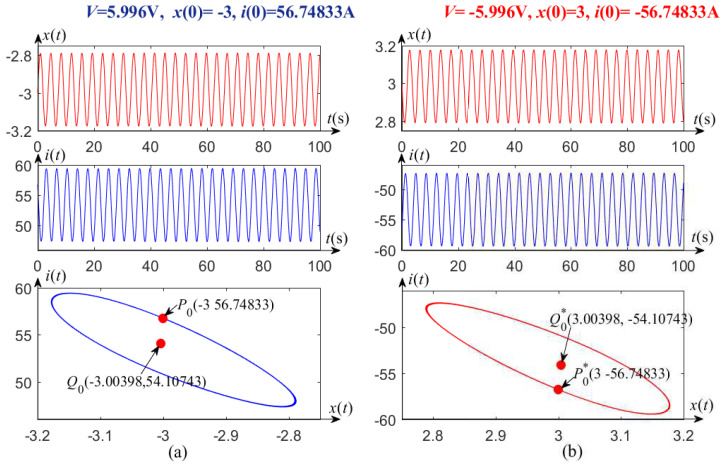
Unstable limit cycles and waveforms of the memristor-based dynamic system, with (**a**) V=5.996 V, (**b**) V=−5.996 V.

**Figure 4 entropy-21-00415-f004:**
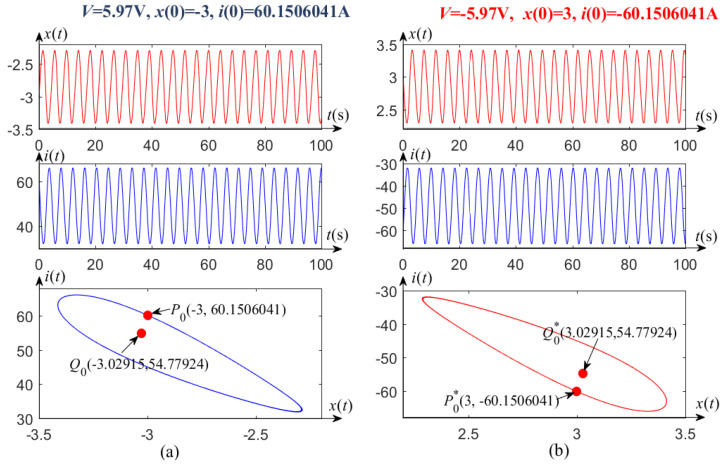
Unstable limit cycles and waveforms of the memristor-based dynamic system, with (**a**) V=5.97 V, (**b**) V=−5.97 V.

**Figure 5 entropy-21-00415-f005:**
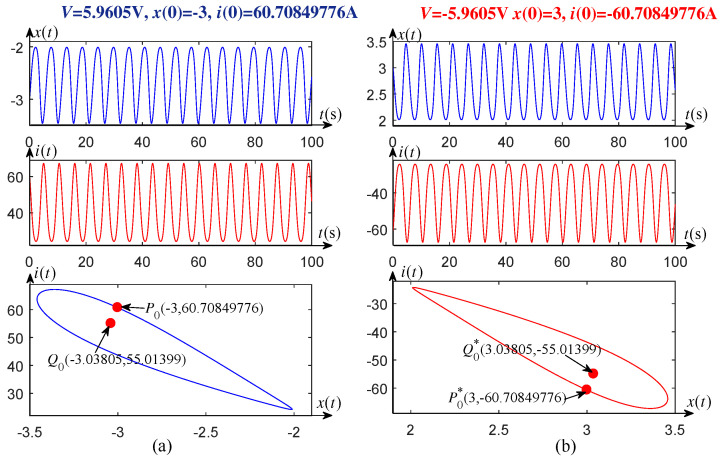
Unstable limit cycles and waveforms of the memristor-based dynamic system, with (**a**) V=5.9605 V, (**b**) V=−5.9605 V.

**Figure 6 entropy-21-00415-f006:**
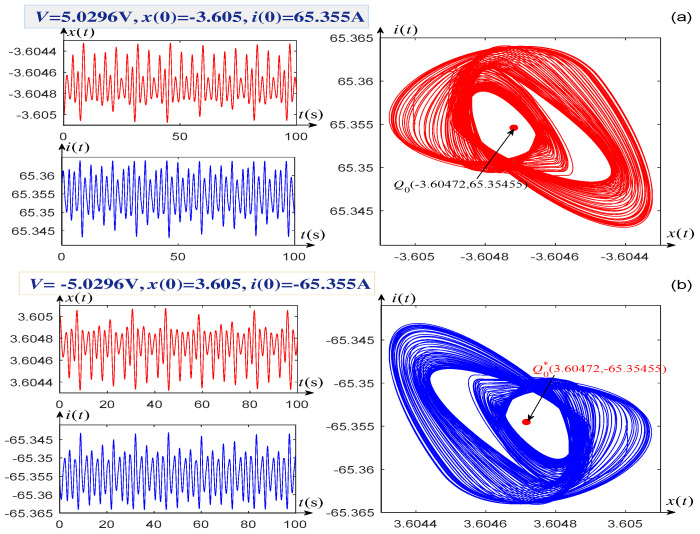
Singular attractors illustrating the complexity of the memrister-based dynamical system (1). (**a**) Singular attractor and waveforms of the state x(t) and current i(t), with V=−5.0296 V and x(0)=−3.605, i(0)=65.355 A; (**b**) Singular attractor and waveforms of x(t) and i(t), with V=5.0296 V and x(0)=3.605, i(0)=−65.355 A.

**Figure 7 entropy-21-00415-f007:**
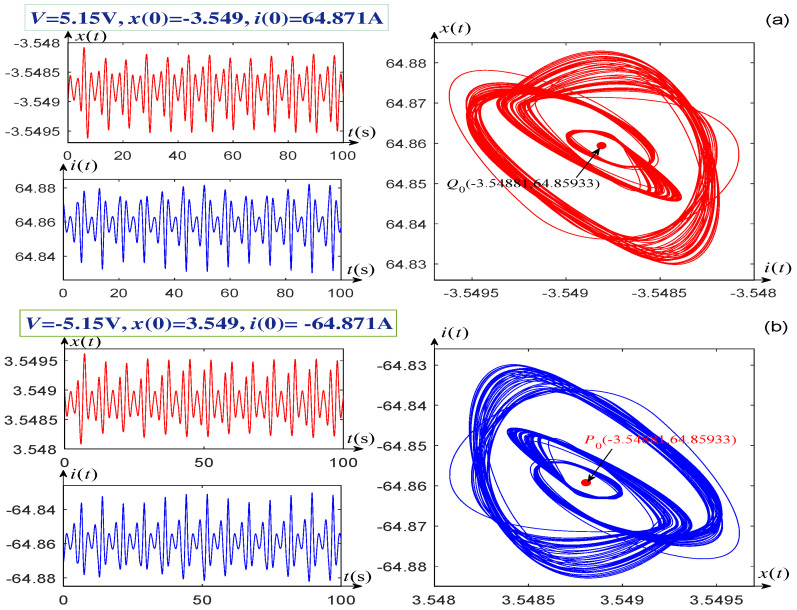
Singular attractors illustrating the complexity of the memrister-based dynamical system (1). (**a**) Singular attractor and waveforms of the state x(t) and current i(t), with V=−5.15 V and x(0)=3.549, i(0)=−64.871 A; (**b**) Singular attractor and wave forms of x(t) and i(t), with V=5.15 V and x(0)=−3.549, i(0)=−64.871 A.

**Figure 8 entropy-21-00415-f008:**
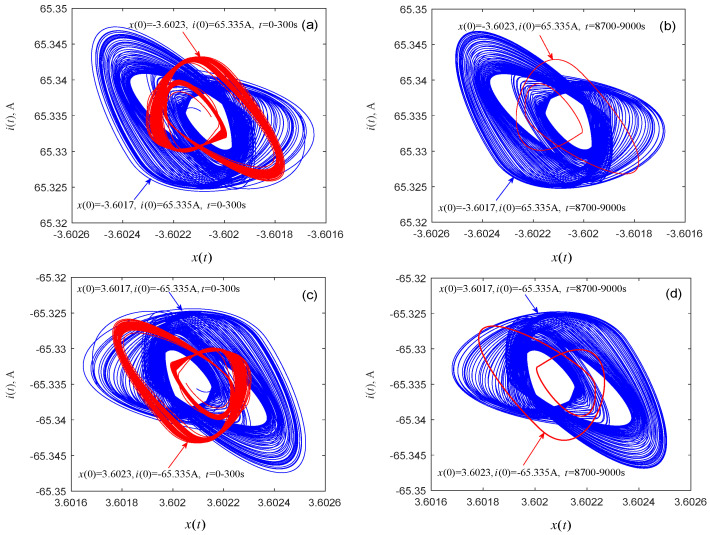
Numerical simulation of the evolution of the coexisting attractors in the memristor-based dynamical system (1). (**a**) V=5.0354 V with duration t= 1–300 s; (**b**) V=5.0354 V with duration *t* = 8700–9000 s; (**c**) V=−5.0354 V with duration *t* = 1–300 s; (**d**) V=−5.0354 V with duration *t* = 8700–9000 s.

**Figure 9 entropy-21-00415-f009:**
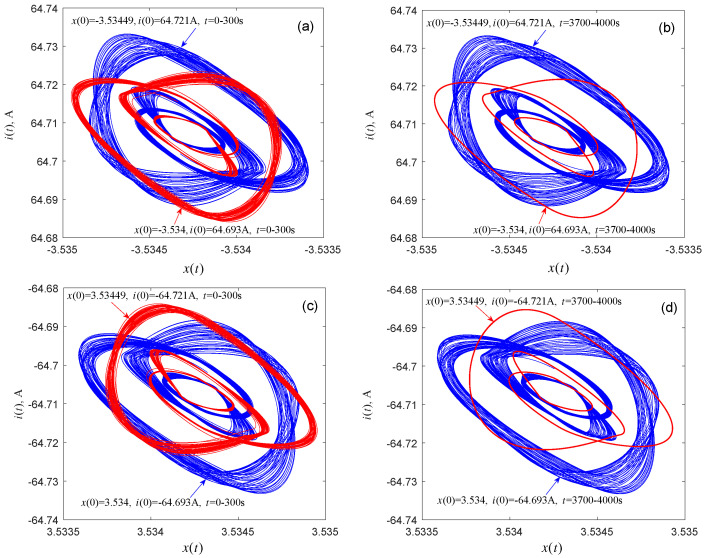
Numerical simulation of the evolution of the coexisting attractors in the memristor-based dynamical system (1). (**a**) V=5.18027 V with duration *t* = 1–300 s; (**b**) V=5.18027 V with duration *t* = 3700–4000 s; (**c**) V=−5.18027 V with duration *t* = 1–300 s; (**d**) V=−5.18027 V with duration *t* = 3700–4000 s.

**Table 1 entropy-21-00415-t001:** Twin local-activity domains of the bistable bi-local active memristor.

Local activity domain 1:	4.6875 V <V<6.25 V, 38.4375 A <I<65.91796875 A
Local activity domain 2:	−6.25 V <V<−4.6875 V, −65.91796875 A <I<−38.4375 A

**Table 2 entropy-21-00415-t002:** Subcritical Hopf bifurcation regions of the memristor-based dynamic system.

Hopf bifurcation region 1:	−6 V <V<−5.96018 V, −55.021798885234276 A <I<−54 A
Hopf bifurcation region 2:	5.96018 V <V<6 V, 54 A <I<55.021798885234276 A

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
