# Peer review of "Unstable Limit Cycles and Singular Attractors in a Two-Dimensional Memristor-Based Dynamic System"

_entropy, 2019, doi:10.3390/e21040415_

Round 1

Reviewer 1 Report

The paper discusses an operational scheme for finding unstable limit cycles and explore the complex dynamical behavior in a 2 two-dimensional system consisting of an inductor and a bistable bi-local active memristor.

The paper is carefully written and it is a reasonably good contribution for

publication in Entropy.

1. The Authors should give some more explanations on what they mean by a singular attractor, maybe by also quoting some related literature.

2. An extended memristor is used in the circuit of Fig. 1. It is unclear how 

the Authors have chosen the law dx/dt = -x|x|+5x+v that governs the state

variable dynamics. Some comments are needed.

3. The Authors use a so called bistable bi-local active memristor and refer

to [25]. The Authors should explicitly explain what the adjectives 

bistable bi-local active mean.

4. The inductor is expected in practice to have a parasitic resistance.

Are the attractors robust with respect to such a parasitic

Author Response

Dear Reviewer

We thank you for you valuable comments and suggestions, which considerably help improve the quality and presentation of our paper.

In the following, we provide our point-to-point responses to the questions, which we hope would be satisfactory.

Thank you for everything.

Best regards

All Authors

Point 1: The Authors should give some more explanations on what they mean by a singular attractor, maybe by also quoting some related literature

Response 1: Singular attractors mentioned in this paper are not those familiar periodic attractors, nor chaotic attractors, but have bizarre quasi-periodic structures (with characteristics similar to chaos). Theoretically, such attractors should be classified as quasi-periodic attractors, because chaotic attractors cannot emerge from a two-dimensional autonomous continuous-time system according to the Poincare-Bendixon theorem. Therefore, in the revision, “singular attractors” are replaced by “singular quasi-periodic attractors” to avoid confusion, where the singularity refers to chaotic-like structures.

Point 2: An extended memristor is used in the circuit of Fig. 1. It is unclear how the Authors have chosen the law dx/dt = -x|x|+5x+v that governs the state variable dynamics. Some comments are needed.

Response 2: The selection of the law dx/dt = -x|x|+5x+v governing the state variable dynamics is inspired by the PWL memristor model dx/dt = 30-x+|x-20|-|x-40|+v proposed by Chua in [2, 3], which has a Shoelace DC V-I Plot and a negative slope region for -10 < V < -3.334, where the capital letter V denotes DC voltage. A phrase “inspired by [3, 22]” has been inserted to the description above Fig. 1 on page 2.

Question 3

The Authors use a so-called bistable bi-local active memristor and refer to [25]. The Authors should explicitly explain what the adjectives bistable bi-local active mean.

Response 3: The following gives a detailed explanation, which has been outlined in the revision (see top of page 2):

(1)   Here, bistability means that the memristor has two different stable solutions originating from both sides of the critical value of the initial state under the same input control.

(2)   The bi-local activity means that the memristor has two bi-local active domains, i.e. 4.6875V< V < 6.25V and - 6.25V< V < - 4.6875V (see [25] for more details).

(3)   Memristors, with two distinct stable pinched hysteresis loops and two local activity domains, are called bistable bi-local-active memristors (see [25] for more details).

Question 4

The inductor is expected in practice to have a parasitic resistance. Are the attractors robust with respect to such a parasitic resistance?

Response 4: The inductor not only has a parasitic resistance but also has a parasitic capacitance in practice. These situations need to be considered in practical circuits design. Since the discussion of this topic is based on the ideal state of inductor, the influence of the parasitic resistance on the robustness of the system attractors has not been studied. However, this question deserves further exploration in the near future.

Reviewer 2 Report

An interesting paper and well written. The scheme can be useful for study memristive dynamical systems.

Author Response

Dear Reviewer

We appreciate your support for our work and your recognition will provide us with continuous driving force for further research and exploration

Thank you for everything.

Best regards

All Authors

Reviewer 3 Report

Small but interesting contribution to nonlinear dynamics.

Author Response

Dear Reviewer,

We appreciate your support for our work and your recognition will provide us with continuous driving force for further research and exploration

Thank you for everything.

Best regards,

All Authors